# Impact and Mechanism of Digital Information Selection on Farmers' Ecological Production Technology Adoption: A Study on Wheat Farmers in China

Yanzi Li [1], Jiahui Xu [1], Fuqiang Liu [2] and Xinshi Zhang [3,*]

1. International Education College, Hebei Finance University, Baoding 071051, China; liyanzi0201@126.com (Y.L.); xujiahui2016106@163.com (J.X.)
2. Baoding Academy of Agricultural Sciences, Baoding 071000, China; liufuq88@163.com
3. Institute of Agricultural Information and Economics, Hebei Academy of Agricultural and Forestry Sciences, Shijiazhuang 050051, China
* Correspondence: zhangXinshi@126.com

**Abstract:** The application of ecological techniques by farmers is important for ensuring the environmentally sustainable advancement of the grain sector. Based on micro-level survey data from 921 Chinese wheat growers in the Hebei and Henan provinces, this study employed an endogenous switching probit model and counterfactual analysis to investigate the impact and mechanisms of digital information utilization on ecological production technology adoption. The results indicated that 43.87% of sample wheat farmers had a low level of adoption of ecological techniques. The utilization of digital information significantly promoted farmers' adoption. If farmers who currently used digital information were to opt-out, the probability of their high adoption would decrease by 11.26%. The utilization of digital information significantly enhanced the adoption of ecological technologies through three mediating factors: technological cognition, production monitoring, and market channels. Therefore, it is imperative to encourage farmers to broaden their social networks and enhance their perception of the importance of digital information. Additionally, it is essential to promote the industrialization and scale operation of wheat production, direct policy subsidies towards new types of management entities, and ensure the accuracy of the supply of digital information for green production through multiple channels. Therefore, it is imperative to expand farmers' social networks and leverage rural communities to increase their perceived importance of digital information. Governments should increase subsidies and promote the scale and industrialization of wheat production. Moreover, the accuracy of digital information supply for sustainable production should be promoted through digital learning platforms, production monitoring systems, and e-commerce networks.

**Keywords:** ecological production technology adoption; digital information utilization; wheat farmers; endogenous switching probit model

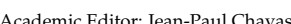



## 1. Introduction

The premium-grade advancement of the grain sector will inevitably entail a sustainable and systematic development that considers both resource and environmental aspects. This signifies a necessary transition of agricultural production towards a greener paradigm [1,2]. As a technological support, ecological production technologies concurrently address resource utilization and ecological governance [3,4]. The Chinese No. 1 Central Document of 2024 emphasizes the need to accelerate the application of techniques for reducing agricultural inputs and increasing efficiency, aiming to facilitate the ecological development of the grain sector by advancing a production technology system guided by environmental considerations. However, farmers face various challenges in actual production, such as technological barriers, financial constraints, and weak risk-bearing

capacity, resulting in a generally low adoption of ecological technologies [5,6]. Therefore, investigating how to promote farmers' adoption and propel ecological transformation has garnered significant attention from both academic and governmental spheres [7–9].

Ecological technology adoption is influenced by many factors, including producer and family characteristics, operational scale, and government subsidies [5,10,11]. Due to the complexity of ecological agricultural technologies, Campenhout [12] found that information acquisition and cognitive abilities affected the decision-making process of rice farmers on ecological production technique adoption. Olabisi et al. [13] and Hong et al. [14] uncovered that technology adoption was significantly and positively influenced by scientifically effective technological information. Digital rural construction facilitates e-commerce and smart agriculture, advancing the digitization process in rural areas [15]. Digital information, conveyed through the Internet using diverse formats such as images and videos, facilitates interaction and the sharing of information [16]. Khan et al. [17] and Zheng et al. [18] suggested that digital information related to ecological agriculture, such as technology training, sales, and rural finance, enhanced farmers' adoption of ecological techniques. However, farmers, as information-disadvantaged groups, often suffer from information discrimination ability deficiency, which refers to their inability to effectively distinguish between true and false information or objective and biased content due to limited knowledge or the influence of social factors. This, coupled with their experience of information cocoons—the selective exposure, absorption, and retention of information based on personal interests, preferences, and habits—creates a self-imposed boundary that restricts their exposure to diverse perspectives. These limitations hinder their ability to adopt ecological technologies [19–21].

While previous literature has presented a valuable analysis of the interplay between digital information utilization and ecological production technology adoption, there is still room for improvement and expansion. Firstly, previous studies on ecological agricultural technology adoption by farmers employed multiple regression or propensity score matching methods, overlooking selection bias caused by unobservable variables in the decision-making process for acquiring digital information. Secondly, existing research often focuses on the accessibility of digital information by employing mobile phones or computer adoption as the proxy. However, the real impact on farmer behavior lies in discerning and utilizing digital information, namely, whether farmers can grasp effective information for ecological production. Thirdly, although some scholars have studied how the Internet impacts ecological technology adoption, there is limited literature focusing on wheat farmers. Wheat, compared to other categories of grain, holds a significant share in acreage and yield in China, making it highly susceptible to natural and market conditions [22]. Therefore, it is particularly necessary to investigate the impact and mechanisms of digital information utilization on the adoption of ecological production technologies among wheat farmers.

In light of the above considerations, the primary purposes of this study were as follows: firstly, to employ an endogenous switching probit model to address inherent endogeneity issues; secondly, based on surveys conducted among wheat growers in Hebei and Henan provinces, to explore how digital information utilization affects ecological production technology adoption by wheat farmers; lastly, to delve into the mechanisms through which digital information influences the adoption of ecological agricultural technologies.

## 2. Theoretical Analysis and Research Hypotheses

Ecological production technologies effectively enhance the standard of goods and improve the ecological surroundings. However, their adoption is related to increased labor, high financial costs, long implementation periods, susceptibility to the external environment, and highly uncertain outcomes [23,24]. Thus, whether to adopt ecological technologies is a typical decision-making process of uncertain outcomes. This uncertainty arises from a dearth of effective information and hinders the anticipation of the probability of events unfolding [25].

The Internet is a pivotal channel for acquiring digital information. It not only dismantles informational barriers, enabling farmers to promptly access relevant information on ecological agricultural technologies, meteorological disasters, and market prices, but also reduces costs associated with information acquisition [26]. Therefore, digital information utilization facilitates ecological agricultural technology adoption by farmers. Hence, Hypothesis 1 was put forward:

**H1.** *Digital information utilization significantly and positively influences ecological production technology adoption by wheat farmers.*

Firstly, Internet information channels provide extensive support for ecological agricultural technologies to farmers, enhancing their awareness of related technologies. The adoption process for farmers for ecological agricultural technologies is categorized into five stages: understanding, interest, evaluation, experimentation, and adoption [27]. A thorough understanding of ecological agricultural technologies is the foundation for farmers to make adoption decisions [28]. Digital information, through diverse presentation methods and interactive communication pathways, stimulates the promotion of ecological technology, meeting the various needs of farmers [29]. Additionally, it establishes an online service platform between farmers and experts. This platform enables farmers to access the latest ecological agricultural technologies and production skills, thereby improving their operational capabilities and proficiency [30]. Therefore, Hypothesis 2 was as follows:

**H2.** *Digital information significantly promotes ecological production technology adoption by stimulating technological cognition.*

Secondly, Internet information channels create social platforms for farmers, thereby fostering online agricultural communities and promoting the application of ecological agricultural technologies. Technology adoption behavior is not solely dependent on farmers' individual choices but is also influenced by social networks based on kinship and geography [31]. However, the high repetition rate of information in these social networks somewhat inhibits the diffusion effectiveness of technology [32]. Digital information extends social relationships based on geographical and familial relationships, allowing farmers to form technical communities by participating in agricultural community discussions. This enables them to conveniently exchange experiences and collaboratively explore methods of ecological production technology, enhancing farmers' learning efficiency and benefiting ecological production [18,33]. Thus, Hypothesis 3 was proposed as below:

**H3.** *Digital information utilization significantly promotes ecological production technology adoption by encouraging information sharing.*

Thirdly, Internet information channels enable the remote monitoring of production, thus allowing farmers to optimize decisions and reducing the uncertainty associated with ecological production technology adoption. Digital information, integrated with navigation technology and sensor technology, has made precision agriculture a reality through the application of information management systems such as automatic weather stations, wheat fusarium warning systems, and smart greenhouses [34–36]. Farmers can use remote monitoring software to track the growth of wheat and critical indicators of the external environment, such as soil quality, meteorological conditions, and pests and diseases. Then, they can accordingly adjust irrigation and fertilization, thus reducing their mistrust and uncertainty of new technologies [37–39]. Hence, this study formulated the following hypothesis:

**H4.** *Digital information utilization significantly promotes ecological production technology adoption by strengthening production monitoring.*

Fourthly, Internet information channels facilitate the connection between farmers and the market, achieving premium prices for green agricultural products and enhancing farmers' expectations of revenues. Compared to ordinary agricultural products, the high production costs of green products have been a key factor limiting farmers' adoption [40]. The Internet and traceable digital technology enable the comprehensive monitoring of the entire process from wheat selection, seedling cultivation, and fertilization to pesticide spraying, which ensures the quality of green agricultural products [41]. Additionally, growers can quickly access information on market demand and price trends through Internet information channels. By leveraging e-commerce, farmers can form better production and sales strategies, enhance consumer trust, and achieve premium prices for high-quality products [42,43]. Digital information creates a more favorable market environment for green agricultural products, motivating farmers to adopt ecological techniques for economic benefits [44]. Therefore, Hypothesis 5 was proposed as below:

**H5.** *Digital information utilization significantly promotes ecological production technology adoption by expanding market channels.*

**H6.** *Digital information utilization significantly promotes ecological production technology adoption by optimizing product traceability.*

Finally, Internet information channels provide convenient financial services for farmers, alleviating the economic pressure associated with ecological production technology adoption. Ecological production technologies are related to high initial investment, and farmers often lack sufficient funds. The high borrowing threshold of traditional financial institutions reduces the accessibility of loans for farmers and limits their adoption of ecological techniques [45]. Under this circumstance, the Internet provides farmers with timely information on credit products, gradually improving farmers' mindsets regarding financial loan services. Financial institutions, leveraging the Internet, continuously improve their online credit rating and credit granting systems, forming a full-process service for online borrowing and payment. This provides flexible financial support for farmers in ecological production, alleviating the financial pressure associated with ecological production technology adoption [45,46]. Furthermore, technique adoption by farmers is susceptible to losses due to market fluctuations and improper technology utilization. Nevertheless, the procedures for traditional agricultural insurance are intricate, and the compensation amounts are modest. Digital information strengthens the risk-sharing network for farmers, offering specialized insurance products to address the impact of risks and effectively alleviate the financial pressure associated with ecological production for farmers [47,48].

**H7.** *Digital information utilization significantly promotes ecological production technology adoption by supporting financial services.*

In summary, digital information significantly influences farmers' adoption of ecological agricultural technologies by promoting technological cognition, information sharing, production monitoring, market channels, product traceability, and financial services, as illustrated in Figure 1.

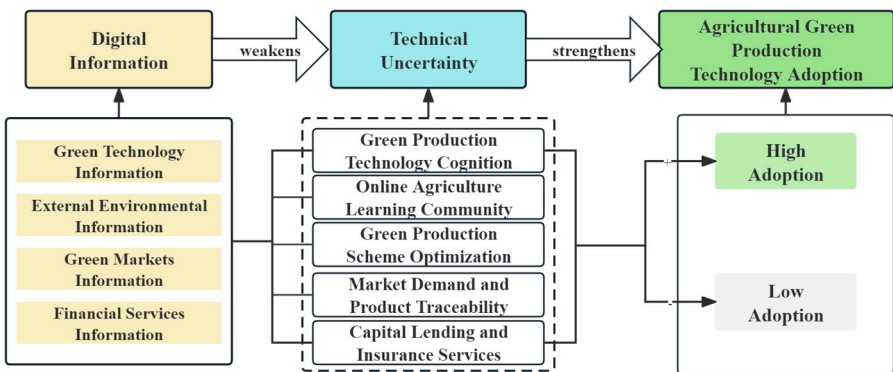

**Figure 1.** Theoretical framework.

## 3. Materials and Methods

### 3.1. Data Collection

The data were collected through household surveys conducted in Hebei and Henan provinces in China between June and July 2023. These two provinces are significant wheat-producing areas in China, boasting a large number of wheat farmers with a wide range of farm sizes and business models. Their varied climate and soil conditions also reflect those found across China's primary wheat-growing regions. They were chosen as the sample areas due to Henan's status as the largest wheat-planting province in China, leading in planting area, annual output value, and number of wheat farmers. Hebei also holds a prominent position, consistently ranking among the top four in wheat planting area and yield, with the highest number of wheat processing enterprises in the country.

The survey adopted a method combining hierarchical and random sampling. In Henan and Hebei provinces, five counties, including Gaocheng, Yuanshi, Neiqiu, Yuzhou, and Wancheng, were randomly selected, comprising 22 sample towns and 67 sample villages, totaling 936 households. The survey questionnaire was filled out through face-to-face interviews, covering topics on basic personal and family information, production and operation situations, utilization status of digital information, and adoption status of ecological production techniques. The research excluded 15 invalid questionnaires, leaving 921 valid questionnaires collected in total. (After excluding 15 invalid questionnaires due to discrepancies with actual circumstances, contradictions among answers, or nearly a quarter of the questions being left unanswered, the research was left with a total of 921 valid questionnaires).

### 3.2. Methodology

Due to potential endogeneity and selection bias issues, this study employed an endogenous switching probit model to estimate how digital information utilization influences wheat growers' adoption of ecological production technologies. Based on this model, counterfactual analysis was used to estimate the treatment effects of digital information utilization on the probability of high adoption among farmers. Stata 15.0 (StataCorp LLC., College Station, TX, USA) was used to perfom the analysis in this study.

#### 3.2.1. Endogenous Switching Probit (ESP) Model

Equation (1) investigated the factors influencing the digital information utilization of wheat farmers.

$$G_i^* = \gamma Z_i + \mu_i$$
$$\text{If } G_i^* > 0, \text{ then } G_i = 1; \text{ else } G_i = 0 \tag{1}$$

$G_i^*$ represents the latent variable indicating the propensity of households to utilize digital information, while $G_i$ indicates the observed situation of digital information utilization. $G_i = 1$ indicates that farmer i has chosen to use digital information, while $G_i = 0$ means farmer i has not. The vector $Z_i$ represents the independent variables affecting digital infor-

mation utilization, including wheat farmers' personal characteristics, family characteristics, and policy environment. $\mu_i$ is the random error term.

To estimate the impact of digital information on wheat farmers' adoption of ecological production technologies, this study constructed Equation (2) as below.

$$R_i = \beta_i X_i + \delta G_i + \varepsilon_i \tag{2}$$

$R_i$ is the dependent variable and represents the adoption of ecological production technologies by wheat farmers. $X_i$ represents control variables and is included in $Z_i$, which contains at least one additional instrumental variable. The instrumental variable needed to meet the conditions of exogeneity and relevance, meaning it needed to correlate with the utilization of digital information but not to ecological production technology adoption. The term $\varepsilon_i$ represents the random error term. Since wheat farmers' digital information utilization decisions may be influenced by unobservable factors, which, in turn, could be related to $R_i$, $G_i$ in Equation (2) could correlate with $\varepsilon_i$, leading to estimation bias due to sample selection issues when directly estimating Equation (2).

Then, Equation (3) determined the adoption of techniques among wheat growers in scenarios where digital information was used and not used.

$$
\begin{aligned}
R_{Mi} &= \beta_M X_{Mi} + \sigma_{Mu}\lambda_{Mi} + \varepsilon_{Mi}, \ if \ G_i = 1 \\
R_{Ni} &= \beta_N X_{Ni} + \sigma_{Nu}\lambda_{Ni} + \varepsilon_{Ni}, \quad if \ G_i = 0
\end{aligned}
\tag{3}
$$

$R_{Mi}$ and $R_{Ni}$ represent the effects of using and not using digital information on ecological production technology adoption, respectively. $X_{Mi}$ and $X_{Ni}$ represent factors affecting the adoption of ecological production technologies among the two groups of wheat farmers. $\varepsilon_{Mi}$ and $\varepsilon_{Ni}$ are random error terms. $\lambda_{Mi}$ and $\lambda_{Ni}$ represent the selection bias caused by unobservable factors, respectively. $\sigma_{Mu} = \text{cov}(\mu_i, \varepsilon_{Mi})$ and $\sigma_{Nu} = \text{cov}(\mu_i, \varepsilon_{Ni})$ denote the covariance between the error terms of the equation of digital information utilization and the determination equation of ecological production technology adoption, respectively. The standardization process yielded the variables $\rho_{Mu}$ and $\rho_{Nu}$ from $\sigma_{Mu}$ and $\sigma_{Nu}$, where $\rho_{Mu}$ represents the coefficient between $\mu_i$ and $\varepsilon_{Ni}$ and $\rho_{Nu}$ represents the coefficient between $\mu_i$ and $\varepsilon_{Ni}$. If $\rho_{Mu}$ and $\rho_{Nu}$ were statistically significant, this indicated that there was selection bias in the model for wheat farmers' ecological production technology adoption.

Finally, by combining Equations (1) and (3) into a system of equations and using maximum likelihood estimation, we obtained the estimated values of the parameters in the models [49,50].

### 3.2.2. Average Treatment Effect

To estimate the average treatment effect on the treated (ATT) of digital information utilization for wheat farmers, this study employed the counterfactual analysis method to compare the adoption levels of ecological production technology between households who used and did not use digital information [51]. Based on Equation (3) of the endogenous switching probit model, the probability of high adoption among wheat growers who used digital information was given by Equation (4).

$$E[R_{Mi}|G_i = 1] = \beta_M X_{Mi} + \sigma_{Mu}\lambda_{Mi} \tag{4}$$

Under the counterfactual assumption, if wheat farmers who utilized digital information did not use it, the probability of high adoption was calculated by Equation (4).

$$E[R_{Ni}|G_i = 1] = \beta_N X_{Mi} + \sigma_{Nu}\lambda_{Mi} \tag{5}$$

By comparing Equations (4) and (5), the ATT of high adoption of ecological production technology among households utilizing digital information could be expressed as Equation (6).

$$ATT \; = \; E[R_{Mi}\,|\,G_i \; = \; 1] - E[R_{Ni}\,|\,G_i \; = \; 1] \; = \; (\beta_M - \beta_N)X_{Mi} - (\sigma_{Mu} - \sigma_{Nu})\lambda_{Mi} \quad (6)$$

3.2.3. Mediating Effect Model

This followed the mediating models of Rucke et al. [52] and Huber [53], referring to the research by Heckman et al. [54], to quantitatively analyze the role of influencing mechanisms on digital information utilization and ecological production technology adoption. Equations (7)–(9) were constructed as below.

$$Y_i \; = \; c_1 G_i + \beta_1 Z_i + \varepsilon_1 \quad (7)$$

$$M_i \; = \; aG_i + \beta_2 Z_i + \varepsilon_2 \quad (8)$$

$$Y_i \; = \; c_2 G_i + bM_i + \beta_3 Z_i + \varepsilon_3 \quad (9)$$

$Y_i$ represents ecological production technology adoption while $G_i$ denotes digital information utilization. $M_i$ and $Z_i$ represent mediating variables and control variables, respectively. $\varepsilon_1 \sim \varepsilon_3$ are the random terms. The coefficients $a$, $b$, $c_1$, $c_2$, $\beta_1$, $\beta_2$, and $\beta_3$ were estimated by ordinary least squares (OLS) regressions to quantify the mediating effects.

*3.3. Variable Definition and Description*

3.3.1. Dependent Variable

Ecological production technology adoption among wheat farmers was the dependent variable. Referencing Ghadiyali et al. [23], Khan et al. [55], and Sun et al. [56] and following the 'Guidelines for Agricultural Green Development Technologies (2018–2030)' released by the Chinese Ministry of Agriculture and Rural Affairs, this study selected five wheat ecological technologies for the key stages of pre-planting and mid-planting for wheat ecological production: superior species selecting and breeding, subsoiling and tillage, water-saving irrigation, soil testing and formulated fertilization, and green control of pests and diseases. Due to significant differences in attributes and effects among different ecological production technologies, following Chen et al. [57] and Wang et al. [58], a five-level Likert scale was used to weigh each technology based on three dimensions: the effect of yield increase per unit area, the degree of improvement in land quality, and the enhancement of the ecological environment. In the field survey of farmers, questionnaire options included 'adopt' and 'not adopt', assigned values of 1 and 0, respectively. The weighted average of five ecological production technologies measured the degree of adoption. To more intuitively determine the high or low adoption level of wheat farmers for ecological production technologies, following the study by Rocha et al. [59], if the level of adoption was greater than or equal to 0.5, it was considered 'high adoption', while an adoption level less than 0.5 was considered 'low adoption'.

The coefficient of variation method assigns weights to indicators by measuring each indicator's degree of variation in a sample [58]. If a technology had a greater degree of variation, its promotion would be more difficult. Therefore, such technology needed to be assigned a higher weight, making it the focus of later ecological production technology promotion [54]. The following process assigned weights under the coefficient of variation method.

Firstly, Equation (10) calculated the coefficient of variation of the five techniques across three dimensions: the effect of yield increase per unit area, the degree of improvement in land quality, and the enhancement of the ecological environment.

$$V_{rj} \; = \; \frac{\sigma_{rj}}{A_{rj}}, r \; = \; 1, 2 \ldots 5; \; j \; = \; 1, 2, 3 \quad (10)$$

In Equation (10), $V_{rj}$ represents the coefficient of variation of the rth technology in the jth dimension. $\sigma_{rj}$ and $A_{rj}$ denote the corresponding standard deviation and mean value, respectively.

Secondly, Equation (11) calculated the secondary index weights of each technology through normalization.

$$W_{rj} = \frac{V_{rj}}{\sum_{r=1}^{5} \sum_{j=1}^{3} V_{rj}} \tag{11}$$

Finally, the primary index weight of $W_r$ in Equation (12) was obtained by summing up the secondary index weights of each ecological production technology across the three dimensions.

$$W_r = \sum_{j=1}^{3} W_{rj} \tag{12}$$

Table 1 ranks the weight of primary indicators for each ecological production technology from highest to lowest: superior species selection and breeding, water-saving irrigation, soil testing and formulated fertilization, subsoiling and tillage, and green control of pests and diseases.

**Table 1.** The weighting of ecological production technology adoption.

| Primary Indicator | Secondary Indicator | Coefficient of Variation | Weight of Secondary Indicator | Weight of Primary Indicator |
|---|---|---|---|---|
| Superior species selecting and breeding | Yield increase per unit area | 0.645 | 0.126 | 0.257 |
| | Improvement in land quality | 0.346 | 0.065 | |
| | Enhancement of ecological environment | 0.328 | 0.064 | |
| Subsoilling and tillage | Yield increase per unit area | 0.275 | 0.054 | 0.189 |
| | Improvement in land quality | 0.358 | 0.070 | |
| | Enhancement of ecological environment | 0.337 | 0.066 | |
| Water-saving irrigation | Yield increase per unit area | 0.267 | 0.052 | 0.204 |
| | Improvement in land quality | 0.233 | 0.045 | |
| | Enhancement of ecological environment | 0.547 | 0.107 | |
| Soil testing and formulated fertilization | Yield increase per unit area | 0.279 | 0.054 | 0.191 |
| | Improvement in land quality | 0.232 | 0.045 | |
| | Enhancement of ecological environment | 0.466 | 0.091 | |
| Green control of pests and diseases | Yield increase per unit area | 0.284 | 0.055 | 0.159 |
| | Improvement in land quality | 0.239 | 0.047 | |
| | Enhancement of ecological environment | 0.292 | 0.057 | |

### 3.3.2. Independent Variable

Digital information utilization was considered the independent variable of this research. The following question was given in the survey: Do you use the Internet to acquire information on ecological techniques, for example, variety selection, scientific fertilization, water-saving irrigation, field management, and pest and disease control? If the response was affirmative, indicating the use of digital information, the variable was assigned a value of 1. Otherwise, if the answer was negative, the variable was assigned a value of 0.

According to the statistical results in Table 2, 47.12% of the surveyed farmers chose the Internet to acquire digital information on ecological production technologies. However, it was found that nearly half of the farmers from the main wheat planting areas in China still have low adoption. Among the farmers who use digital information, the proportion of high adoption of ecological technologies reached 57.64%, which was much higher than

the proportion of low adoption. On the other hand, among the farmers who did not use digital information, the proportion of low adoption of ecological production technologies was more than half, which was 22.47% higher than the proportion of the whole sample. Therefore, it can be inferred that digital information utilization by farmers increases the adoption of ecological techniques.

**Table 2.** Descriptive statistics of wheat farmers' adoption.

| | Observation | Percentage (%) | Low Adoption of Ecological Technologies | | High Adoption of Ecological Technologies | |
|---|---|---|---|---|---|---|
| | | | Observation | Percentage (%) | Observation | Percentage (%) |
| Using digital information | 434 | 47.12 | 136 | 33.66 | 298 | 57.64 |
| Not using digital information | 487 | 52.88 | 268 | 66.34 | 219 | 42.36 |
| Total | 921 | | 404 | 43.87 | 517 | 56.13 |

### 3.3.3. Control Variable

According to Adnan et al. [24], Gao et al. [28], and Zhou et al. [60] and based on practical observations, the following variables were selected to reflect individual characteristics of wheat farmers: gender, age, education level, employment in other work, and serving as village cadres. To reflect family characteristics, the selected variables were household population, household agricultural labor ratio, proportion of non-agricultural income, cultivated land area, and number of cultivated land plots. Moreover, government policy support was indicated by the receipt of government subsidies.

### 3.3.4. Instrumental Variable

The instrumental variable needed to simultaneously satisfy the exogeneity and relevance conditions. Following Zhou et al. [61] and Chandio et al. [62], the perceived importance of digital information served as the instrumental variable. If farmers perceived information from the Internet as valuable, they would access digital information.

To further verify the effectiveness of the instrumental variable, this study initially regressed ecological production technology adoption as the dependent variable against digital information utilization and the perceived importance of digital information as independent variables. The coefficient for the perceived importance of digital information was 0.047, with a $p$-value of 0.649, indicating that it did not directly influence the adoption of ecological technology, thus satisfying the exogeneity requirement. Subsequently, when employing digital information utilization as the dependent variable and the perceived importance of digital information as the independent variable, the coefficient for the perceived importance of digital information was 1.607, with a $p$-value of 0.000, demonstrating its significant impact on farmers' digital information utilization and fulfilling the relevance condition.

### 3.3.5. Mediating Variable

Technological cognition, information sharing, production monitoring, market channels, product traceability, and financial services were mediating variables in this study. Relevant questions were designed based on the situation of the main wheat-producing areas, and five-point Likert scales were used to measure the above indicators. Table 3 shows the definitions and descriptive statistics of the variables.

**Table 3.** Definition and descriptive statistics.

| Variable | Definition | Mean | Standard Variation |
|---|---|---|---|
| Ecological production technology adoption | Calculate by the coefficient of variation method: 1 = High adoption; 0 = Low adoption. | 0.561 | 0.496 |
| Digital information utilization | Do you use the Internet to access information on wheat ecological production technologies such as variety selection, scientific fertilization, water-saving irrigation, farmland management, and pest control? 1 = Yes; 0 = No. | 0.471 | 0.499 |
| Perceived importance of digital information | Perception of the importance of digital information: 1 = Not important; 2 = Average; 3 = Very important. | 1.543 | 0.666 |
| Gender | Female = 0; Male = 1. | 0.713 | 0.452 |
| Age | Age of farmers. | 60.98 | 8.259 |
| Educational level | 1 = No formal education; 2 = Elementary school graduate; 3 = Junior high school graduate; 4 = High school graduate or above. | 2.828 | 0.878 |
| Engagement in other work | 1 = Only engaged in grain production; 2 = Migrant work; 3 = Self-employed in grain production; 4 = Other non-agricultural work. | 1.550 | 0.742 |
| Serving as village cadres | 0 = No; 1 = Yes. | 0.137 | 0.344 |
| Household population | Number of household members. | 4.543 | 2.245 |
| Household agricultural labor ratio | Number of household members engaged in farming/Total household labor force (%). | 0.778 | 0.388 |
| Proportion of non-agricultural income | Household non-farm income/Total household income. | 0.674 | 0.331 |
| Cultivated land area | Hectares of managed land (including leased land). | 1.938 | 4.322 |
| Number of cultivated land plots | Number of managed land plots (including leased land). | 3.837 | 2.333 |
| Receipt of government subsidies | If receives government subsidies or compensation after a disaster: Yes = 1; No = 0. | 0.128 | 0.334 |
| Technological cognition | Do you think the Internet, such as Apps, WeChat, Douyin, Kuaishou, etc., helps you master the relevant skills of wheat ecological production and management? 1 = Absolutely not capable; 2 = Not very capable; 3 = Hard to say; 4 = Comparatively capable; 5 = Extremely capable. | 2.817 | 0.918 |
| Information sharing | Do you think the Internet, such as interactive learning communities, expedites the sharing of wheat ecological production experience? 1 = Absolutely not; 2 = No; 3 = Hard to say; 4 = Yes; 5 = Absolutely Yes. | 2.084 | 0.762 |
| Production monitoring | Do you think the Internet, such as smart agriculture platforms, helps to monitor pests and disease, soil moisture, and other ecological production information? 1 = Absolutely not; 2 = No; 3 = Hard to say; 4 = Yes; 5 = Absolutely Yes. | 2.870 | 0.710 |
| Market channels | Do you think the Internet provides you with information about wheat demand and prices and online sales channels? 1 = Absolutely not; 2 = No; 3 = Hard to say; 4 = Yes; 5 = Absolutely Yes. | 2.870 | 1.010 |
| Product traceability | Do you think the Internet helps to promote the process and quality of wheat ecological production through green agricultural product traceability platforms? 1 = Absolutely not; 2 = No; 3 = Hard to say; 4 = Yes; 5 = Absolutely Yes. | 2.031 | 0.800 |
| Financial services | Do you think the Internet provides you with online lending, agricultural insurance, and other financial services? 1 = Absolutely not; 2 = No; 3 = Hard to say; 4 = Yes; 5 = Absolutely Yes. | 3.054 | 2.031 |

## 4. Results and Discussion

### 4.1. Probit Regression Results

The results of the endogenous switching probit models for ecological technique adoption and digital information utilization are illustrated in Table 4. The significant non-zero value of the Wald chi-square test and the significance of rho1 at the 5% level indicated the statistical validity of the model specification and the presence of selection bias. The positive estimate of rho1 suggested that wheat growers using digital information had a higher probability of adopting ecological technologies compared to the average level among the sample.

**Table 4.** Endogenous switching probit regression results (*n* = 921).

| Variable | Digital Information Utilization | Ecological Production Technology Adoption | |
|---|---|---|---|
| | | Used Digital Information | Did Not Use Digital Information |
| Perceived importance of digital information | 1.7694 *** (0.1221) | | |
| Gender | −0.8674 *** (0.1678) | 1.5037 *** (0.4485) | 0.1435 (0.1693) |
| Age | −0.1719 *** (0.0138) | −0.1750 *** (0.0468) | 0.0042 (0.0113) |
| Educational level | 0.5967 (0.0822) | 0.2161 (0.2484) | 0.1470 (0.0942) |
| Engagement in other work | 0.4690 *** (0.1202) | −1.6713 *** (0.5870) | −0.0686 (0.1211) |
| Serving as village cadres | 0.0009 (0.2250) | −0.9409 (0.8014) | 0.5799 *** (0.2444) |
| Household population | 0.0404 (0.0333) | 0.0700 (0.0969) | −0.0219 (0.0340) |
| Household agricultural labor ratio | 0.4067 *** (0.1538) | 0.3017 (0.4422) | 0.8954 *** (0.2828) |
| Proportion of non-agricultural income | −0.0502 (0.2602) | −3.7049 *** (0.9923) | 1.0139 *** (0.2669) |
| Cultivated land area | 0.2146 *** (0.0418) | 9.5217 *** (1.5370) | 0.4469 *** (0.0591) |
| Number of cultivated land plots | −0.1490 *** (0.0322) | −0.0322 (0.1008) | −0.0367 (0.0375) |
| Receipt of government subsidies | −0.1195 (0.2041) | 1.7679 *** (0.5732) | 0.3639 (0.2114) |
| Constant | −12.7713 *** (1.0016) | 6.9450 ** (3.4780) | 1.1541 (0.7946) |
| $\rho_{Mu}$ | | 0.6409 ** (0.2279) | |
| $\rho_{Nu}$ | | | 1 ($6.00 \times 10^{-11}$) |
| Log-likelihood = −500.9205 | | | |
| Wald chi2(12) = 314.83 *** | | | |
| LR test of indep. eqns. (rho1 = rho0 = 0):chi2(2) = 32.63 Prob > chi2 = 0.0000 | | | |

Note: ***, and ** indicate significance levels at 1% and 5%, respectively. Numbers in parentheses are standard errors.

Firstly, this research analyzed factors influencing wheat farmers' utilization of digital information. Gender, age, engagement in other work, cultivated land area, number of cultivated land plots, household agricultural labor ratio, and perceived importance of digital information significantly affected wheat farmers' utilization of digital information. Specifically, female farmers were more inclined to use digital information due to their relatively limited social capital and the need to address information asymmetry through online channels [17]. In contrast, older individuals were less inclined to use digital information as a result of a lack of Internet skills and skepticism towards the authenticity and credibility of digital information, preferring interpersonal communication for information

acquisition [62]. The degree of engagement in other work positively influenced the utilization of digital information because farmers involved in individual businesses or off-farm work needed to establish a broader social network, acquire more market information, and expand online sales channels, and thus rely more on digital information [63]. Additionally, part-time farmers usually had higher incomes, providing an economic basis for their use of digital information. Larger cultivated land areas and fewer plots made wheat farmers more likely to choose digital information because large-scale grain growers incurred higher costs and were more concerned about the application of new techniques, thus preferring to broaden information to gain more opportunities and benefits [64]. Furthermore, the higher the household agricultural labor ratio, the more likely they would employ digital information because with more farming laborers in the household, there was repeated communication among family members about wheat growth conditions, and the convenience of the Internet facilitated efficient communication.

In addition, this study found significant differences in the factors impacting the adoption level between wheat farmers who used digital information and those who did not. Gender, age, engagement in other work, and receipt of government subsidies significantly impacted ecological production technology adoption among wheat farmers who used digital information. This indicated that among wheat farmers who chose digital information, young male farmers who specialized in wheat production had a higher adoption level. Young male farmers usually expanded the wheat production scale and engaged in industrialized operations to enhance economic efficiency [65]. They had a deeper understanding of ecological wheat production along with a higher risk acceptance and greater information acquisition capabilities and were thus more willing to adopt wheat ecological production technologies [66]. Moreover, government subsidies for wheat farmers encouraged them to apply ecological techniques, thereby increasing the adoption level [67].

For farmers who did not use digital information, serving as village cadres and the household agricultural labor ratio significantly affected their adoption of ecological techniques. Farmers who had served as village cadres had a higher adoption. They played an important role in promoting ecological techniques, thus having a positive understanding and high expectation of ecological technologies, which increased the adoption level [68,69]. The lower the household agricultural labor ratio, the more likely farmers were to not apply ecological techniques. Households with fewer farming laborers faced greater pressure in grain production management and market sales, lacked sufficient labor to engage in field ecological management and market sales of wheat, and had limited access to the latest ecological agricultural techniques, consequently hindering the adoption of ecological technologies [70].

Both cultivated land area and the proportion of non-agricultural income significantly influenced ecological production technology adoption among farmers who used or did not use digital information. A larger land area among wheat farmers indicated a stronger economic capacity and greater awareness of ecological production, thus facilitating the adoption of ecological technologies [71]. It was noteworthy that for farmers using digital information, a higher proportion of agricultural income led to higher adoption. They typically engaged in large-scale grain production operations and were more likely to apply ecological techniques to enhance yield and quality, thereby increasing agricultural income [71]. Conversely, a smaller proportion of agricultural income corresponded to higher adoption for non-using farmers. For such growers, they focused more on reducing production costs because agricultural income may not have been their primary economic source. Diverse social relationships may strengthen their awareness of ecological development, thus leading to a willingness to adopt ecological techniques, such as superior species selection and breeding, to lower production costs [72]. However, during the survey, it was found that these farmers rarely adopted ecological production technologies that were technically complex and required a high level of labor input.

## 4.2. Average Treatment Effects

The treatment effect of farmers using digital information on ecological production technology options was calculated by Equation (5), shown in Table 5. Overall, digital information utilization positively impacted ecological production technology adoption ($p <$ 0.01). From the results of ATT, it can be observed that for wheat farmers who had already used digital information, if they were to stop using digital information, the probability of their high adoption would decrease by 11.26%. Thus, Hypothesis 1 was supported.

**Table 5.** Average treatment effect results.

| Ecological Production Information Acquisition | Used Digital Information | Did Not Use Digital Information | Average Treatment Effect on the Treated (ATT) |
|---|---|---|---|
| Ecological production technology adoption | 0.8854 | 0.7857 | 0.0997 *** |

Note: *** indicates significance levels at 1%.

## 4.3. Mediating Effects

This study analyzed the mechanism through which wheat farmers' utilization of digital information influenced ecological production technology adoption using six mediating variables: technological cognition, information sharing, production monitoring, market channels, product traceability, and financial services. The results are presented in Tables 6 and 7.

**Table 6.** Mediating effect results of Technological Cognition, Information Sharing and Production Monitoring variables.

| | Technological Cognition | | | Information Sharing | | | Production Monitoring | | |
|---|---|---|---|---|---|---|---|---|---|
| | Regression (1) Ecological Production Technology Adoption | Regression (2) Technological Cognition | Regression (3) Ecological Production Technology Adoption | Regression (4) Ecological Production Technology Adoption | Regression (5) Information Sharing | Regression (6) Ecological Production Technology Adoption | Regression (7) Ecological Production Technology Adoption | Regression (8) Production Monitoring | Regression (9) Ecological Production Technology Adoption |
| Digital information utilization | 0.1373 *** (0.0383) | 0.3146 *** (0.0734) | 0.0737 (0.0357) | 0.1373 *** (0.0383) | 0.8621 *** (0.0499) | 0.1065 (0.0442) | 0.1373 *** (0.0383) | 0.2533 *** (0.0600) | 0.0734 (0.0356) |
| Technological cognition | | | 0.2023 *** (0.0160) | | | | | | |
| Information sharing | | | | | | 0.0357 (0.0255) | | | |
| Production monitoring | | | | | | | | | 0.2521 *** (0.0195) |

Note: *** indicates significance levels at 1%. Numbers in parentheses are standard errors.

**Table 7.** Mediating effect results of Market Channels, Product Traceability and Financial Services variables.

| | Market Channels | | | Product Traceability | | | Financial Services | | |
|---|---|---|---|---|---|---|---|---|---|
| | Regression (10) Ecological Production Technology Adoption | Regression (11) Market Channels | Regression (12) Ecological Production Technology Adoption | Regression (13) Ecological Production Technology Adoption | Regression (14) Product Traceability | Regression (15) Ecological Production Technology Adoption | Regression (16) Ecological Production Technology Adoption | Regression (17) Financial Services | Regression (18) Ecological Production Technology Adoption |
| Digital information utilization | 0.1373 *** (0.0383) | 0.2789 *** (0.0825) | 0.0897 (0.0358) | 0.1373 *** (0.0383) | 0.8871 *** (0.0540) | 0.1123 (0.0436) | 0.1373 *** (0.0383) | 0.3581 (0.2197) | 0.1332 *** (0.0383) |
| Market channels | | | 0.1742 *** (0.0143) | | | | | | |
| Product traceability | | | | | | 0.02815 (0.0235) | | | |
| Financial services | | | | | | | | | 0.0116 (0.0058) |

Note: *** indicates significance levels at 1%. Numbers in parentheses are standard errors.

From Tables 6 and 7, regression (1) demonstrated a significant positive effect of digital information utilization on ecological technique adoption, further supporting Hypothesis 1. Regressions (2) to (18) indicated that digital information utilization significantly promoted farmers' adoption by enhancing their technological cognition, strengthening production monitoring, and expanding market channels, confirming Hypotheses 2, 4, and 5.

Regression (5) indicated that farmers using digital information significantly promoted information sharing among farmers. However, regression (6) showed that information sharing did not significantly affect ecological technology adoption. Due to economic interests and competitive relationships, farmers are reluctant to share important information about the implementation of wheat ecological production technologies. A lack of trust and cooperation mechanisms among farmers may hinder the improvement of ecological technology adoption [73]. Regression (14) suggested that digital information significantly promoted the traceability of production processes and the promotion of product quality. However, regression (15) indicated that the traceability of green products did not significantly promote ecological production technology adoption. It is difficult to guarantee the authenticity of digital information, leading to consumer distrust of high-quality green wheat products and subsequently reducing wheat ecological production technology adoption [74]. Regression (17) showed that the Internet was not a significant provider of financial services to farmers. Financial institutions maintained a high level of vigilance against credit risks and farmers lacked a robust credit rating system. Additionally, farmers had doubts about the security of Internet financial services due to conservative attitudes [75]. Therefore, digital information did not influence green production technology adoption by encouraging information sharing, optimizing product traceability, and supporting financial services.

*4.4. Robustness Tests*

This study carried out robustness tests by adjusting the independent variable, which measured the adoption of ecological production technologies.

Initially, the threshold distinguishing high and low levels of ecological production technology adoption among wheat farmers was shifted from 0.5 to 0.6. This change meant that if a wheat farmer's adoption level exceeded 0.6, it was classified as 'high adoption'; otherwise, it was considered 'low adoption'. Tables 8 and 9 show the results. With this revised threshold, the average treatment effect on the treated of wheat farmers' digital information utilization for ecological production technology adoption was significantly positive at the 1% level. Under the counterfactual assumption, if the wheat farmers who used digital information stopped using it, their level of adoption would drop by 16.17%, confirming the robustness of the prior findings.

**Table 8.** ESP regression results after changing the threshold of the dependent variable (*n* = 921).

| Variable | Digital Information Utilization | Ecological Production Technology Adoption | |
|---|---|---|---|
| | | Used Digital Information | Did Not Use Digital Information |
| Perceived importance of digital information | 1.6484 *** | | |
| | (0.1280) | | |
| Gender | −0.8022 *** | 1.0842 *** | 0.1716 |
| | (0.1706) | (0.3617) | (0.1603) |
| Age | −0.1661 *** | −0.1473 *** | 0.0076 |
| | (0.0137) | (0.0384) | (0.0134) |
| Educational level | 0.1129 | 0.1272 | 0.08836 |
| | (0.0811) | (0.2039) | (0.0863) |
| Engagement in other work | 0.4551 *** | −0.9298 ** | 0.1628 |
| | (0.1224) | (0.4116) | (0.1115) |
| Serving as village cadres | 0.1898 | 0.4237 | 0.4936 ** |
| | (0.2194) | (0.5311) | (0.2313) |
| Household population | 0.0220 | 0.0647 | 0.0170 |
| | (0.0311) | (0.0881) | (0.0317) |

**Table 8.** *Cont.*

| Variable | Digital Information Utilization | Ecological Production Technology Adoption | |
|---|---|---|---|
| | | **Used Digital Information** | **Did Not Use Digital Information** |
| Household agricultural labor ratio | 0.3853 *** | 0.2808 | 0.6018 ** |
| | (0.1506) | (0.4959) | (0.2542) |
| Proportion of non-agricultural income | −0.1332 | −1.6108 *** | 0.9066 *** |
| | (0.2439) | (0.6491) | (0.2436) |
| Cultivated land area | 0.0926 * | 7.6672 *** | 0.1432 *** |
| | (0.0565) | (1.0829) | (0.0206) |
| Number of cultivated land plots | −0.1139 *** | −0.1241 | −0.0764 *** |
| | (0.0328) | (0.0912) | (0.0281) |
| Receipt of government subsidies | −0.1482 | 1.1854** | 0.1839 |
| | (0.1939) | (0.5165) | (0.2103) |
| Constant | −12.2002 *** | 5.8752 * | −1.6972 ** |
| | (1.0016) | (3.1476) | (0.8306) |
| $\rho_{Mu}$ | | 0.6749 ** | |
| | | (0.2145) | |
| $\rho_{Nu}$ | | | −0.8792 ** |
| | | | (0.1076) |
| Log-likelihood = −559.2867 | | | |
| Wald chi2(12) = 257.04 *** | | | |
| LR test of indep. eqns. (rho1 = rho0 = 0):chi2(2) = 11.42 Prob > chi2 = 0.0033 | | | |

Note: ***, **, and * indicate significance levels at 1%, 5%, and 10%, respectively. Numbers in parentheses are standard errors.

**Table 9.** Average treatment effect results after changing the threshold of the dependent variable.

| Ecological Production Information Acquisition | Used Digital Information | Did Not Use Digital Information | Average Treatment Effect on the Treated (ATT) |
|---|---|---|---|
| Ecological production technology adoption | 0.9028 | 0.7568 | 0.1460 *** |

Note: *** indicates significance levels at 1%.

Furthermore, this study eliminated the weights assigned to different ecological production technologies and re-estimated Equations (1) and (3), as indicated by Tables 10 and 11. The average treatment effect on the treated (ATT) for wheat farmers' digital information utilization for ecological production technology adoption was 0.9166 and 0.7864, respectively, both statistically significant at the 1% level. This indicated that if wheat farmers who utilized digital information had not done so, the likelihood of achieving a high level of ecological production technology adoption would have dropped by 14.20%. These findings further validate the robustness of the primary results in this paper.

**Table 10.** ESP regression results after eliminating the weights of the dependent variable (*n* = 921).

| Variable | Digital Information Utilization | Ecological Production Technology Adoption | |
|---|---|---|---|
| | | **Used Digital Information** | **Did Not Use Digital Information** |
| Perceived importance of digital information | 1.6252 *** | | |
| | (0.1296) | | |
| Gender | −0.8875 *** | 1.6398 *** | 0.1570 |
| | (0.1723) | (0.3414) | (0.1606) |
| Age | −0.1736 *** | −0.1866 *** | 0.0030 |
| | (0.0135) | (0.0405) | (0.0104) |
| Educational level | 0.1151 | 0.4019 ** | 0.1047 |
| | (0.0793) | (0.1989) | (0.0872) |

**Table 10.** *Cont.*

| Variable | Digital Information Utilization | Ecological Production Technology Adoption | |
| --- | --- | --- | --- |
| | | Used Digital Information | Did Not Use Digital Information |
| Engagement in other work | 0.4508 *** | −1.2423 *** | 0.1514 |
| | (0.1302) | (0.3923) | (0.1199) |
| Serving as village cadres | 0.1336 | 0.6186 | 0.4460 ** |
| | (0.2111) | (0.5433) | (0.2233) |
| Household population | 0.0374 | 0.1534 * | 0.0288 |
| | (0.0320) | (0.0824) | (0.0322) |
| Household agricultural labor ratio | 0.4190 *** | 0.2510 | 0.5610 ** |
| | (0.1502) | (0.2250) | (0.2584) |
| Proportion of non-agricultural income | −0.1495 | −2.7568 *** | 1.0818 *** |
| | (0.2554) | (0.7780) | (0.2439) |
| Cultivated land area | 0.1066 *** | 7.9816 *** | 0.1729 *** |
| | (0.0337) | (1.0841) | (0.0231) |
| Number of cultivated land plots | −0.1109 *** | −0.0188 | −0.0805 *** |
| | (0.0330) | (0.0843) | (0.0287) |
| Receipt of government subsidies | −0.0625 | 1.7000 *** | 0.1687 |
| | (0.1983) | (0.4908) | (0.2008) |
| Constant | −12.6561 *** | 8.2567 *** | −1.5671 ** |
| | (0.9803) | (3.0026) | (0.7705) |
| $\rho_{Mu}$ | | 0.8280 ** | |
| | | (0.1514) | |
| $\rho_{Nu}$ | | | −0.9998 |
| | | | (0.0116) |
| Log-likelihood = −550.119 | | | |
| Wald chi2(12) = 297.26 *** | | | |
| LR test of indep. eqns. (rho1 = rho0 = 0):chi2(2) = 28.54 Prob > chi2 = 0.0000 | | | |

Note: ***, **, and * indicate significance levels at 1%, 5%, and 10%, respectively. Numbers in parentheses are standard errors.

**Table 11.** Average treatment effect results after eliminating the weights of the dependent variable.

| Ecological Production Information Acquisition | Used Digital Information | Did Not Use Digital Information | Average Treatment Effect on the Treated (ATT) |
| --- | --- | --- | --- |
| Ecological production technology adoption | 0.9166 | 0.7864 | 0.1302 *** |

Note: *** indicates significance levels at 1%.

## 5. Conclusions

This study, based on survey data from wheat growers in Hebei and Henan provinces, employed an endogenous switching probit model to analyze the impact of farmers' digital information usage on the adoption of ecological production technologies and delved into the underlying mechanisms. Firstly, factors such as engagement in other work, cultivated land area, household agricultural labor ratio, and perceived importance of digital information significantly influenced farmers' digital information utilization. Secondly, differences existed in the factors affecting the adoption of ecological production technologies between farmers who used digital information and those who did not. Thirdly, digital information utilization notably boosted adoption rates among farmers, with a significant decrease in the probability of high adoption if farmers were to opt-out. Finally, the Internet facilitated ecological production technology adoption through mediating factors such as technological cognition, production monitoring, and market channels. In light of these findings, recommendations are proposed to bolster ecological production technology adoption among wheat farmers and propel the sustainable growth of the wheat sector.

Firstly, to encourage farmers to utilize digital information related to ecological production practices, it is crucial to broaden their social networks and boost their perception

of the importance of digital information. Farmers could join agricultural cooperatives to access a broader range of information sources. Rural communities could highlight the benefits of digital technology in agriculture by sharing successful experiences of utilizing digital information in ecological wheat production. Additionally, following the Rural Revitalization Strategy, the government needs to enhance digital communication infrastructure by increasing the number of mobile communication technology (5G) base stations and fiber-optic broadband and Internet of Things facilities, thereby reducing barriers to digital information access for farmers.

Secondly, the government should increase subsidies and promote the scale and industrialization of wheat production. Farmers who utilize digital information are more likely to adopt ecological production technologies when engaging in large-scale, industrialized agriculture and when receiving government subsidies. Thus, policy subsidies, such as land fertility subsidies and straw utilization subsidies, should focus on supporting entities that operate large-scale wheat farming. Ways to scale up include promoting land transfer to specialize in wheat cultivation or using land trusteeship models to increase scalability and efficiency. A more cohesive and streamlined industry should be created by promoting contract farming that integrates wheat production, processing, and sales.

Finally, it is crucial to ensure a reliable and accurate supply of digital information through multiple channels to promote farmers' ecological production technology adoption. This can be achieved by expanding agricultural database resources, launching digital classrooms, and creating online agricultural learning communities to connect experts with farmers. Then, the precise monitoring of ecological wheat production should be strengthened by advancements in agricultural sensors and big data technology. In addition, e-commerce platforms should be leveraged to provide market information and expand sales channels to help farmers make better-informed decisions, driving the adoption of ecological production technologies.

**Author Contributions:** Conceptualization, X.Z. and Y.L.; methodology, Y.L.; software, J.X.; validation, F.L. and X.Z.; formal analysis, X.Z.; investigation, Y.L., J.X. and F.L.; data curation, F.L.; writing—original draft preparation, Y.L.; writing—review and editing, J.X. and X.Z.; supervision, X.Z.; project administration, X.Z. and Y.L.; funding acquisition, X.Z. and Y.L. All authors have read and agreed to the published version of the manuscript.

**Funding:** This research was funded by the Hebei Agriculture Research System (HBCT2023010301) and the Social Science Foundation of Hebei Province (HB22XW005).

**Institutional Review Board Statement:** Not applicable.

**Data Availability Statement:** The original contributions presented in this study are included in the article; further inquiries can be directed to the corresponding author.

**Acknowledgments:** Special thanks are given to the wheat farmers who were eager to cooperate in the survey.

**Conflicts of Interest:** The authors declare that this research was conducted without any commercial or financial relationships that could be construed as potential conflicts of interest.

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
