# Peer review of "Impact and Mechanism of Digital Information Selection on Farmers’ Ecological Production Technology Adoption: A Study on Wheat Farmers in China"

_agriculture, doi:10.3390/agriculture14050713_

Round 1

Reviewer 1 Report

Comments and Suggestions for Authors

Referee Report for “Impact and Mechanism of Digital Information Selection on 2 Farmers' Green Production Technology Adoption: A Study on Wheat Farmers in China” (Agriculture 2945333)

Summary and Major Comment:

The paper is generally well-written and edited, the dataset is new and novel, and the empirical analysis is competent.  The subject area of the paper represents a contribution to the microeconomic development literature on learning and technology.  There are good and appropriate citations and linkages to the literature throughout the paper.  The data sample is large enough for this type of statistical analysis, is new and novel, and attrition is relatively low. The empirical method appropriately allows differential impacts for digital information use and non-use groups.

Detailed Comments:

Abstract: The authors should consider revising the last two lines of the abstract to better link to what is shown in the paper.  The policy recommendation about investing in infrastructure networks is only an indirect link to the presented model on the dynamics of learning via technology for example.  Please also see expanded comments below. 

1. Introduction

p. 2 lines 55-56, is there a definition of “information discrimination ability deficiency and information silos” that could be added; I am unfamiliar with these terms and wonder if other readers will feel similarly.

2. Theoretical Analysis and Research Hypotheses

The hypotheses are well motivated in terms of links to behavior, learning, information sharing, production efficiencies, and financial markets respectively.

3. Material and Methods

It would be helpful to footnote why the 15 surveys were determined to be “invalid” in order to help the reader better understand.

Instrumental variables are mentioned in line 210 of page 5 but these variables are not specified until much later in the paper.  It is important in the methodology to describe these variables and how and why they satisfy the requirements for valid IVs in this section from a theoretical perspective with the empirical tests in the later section as noted alongside interpretation of results.

In equations (3) and (4), you indicate that G_i*>0 for both cases.  Is this not also true by definition for all cases?  Why do you need this restriction in parentheses here?  If this is referring to some kind of negativity in the probability of digital information use that comes out in the empirics, then this needs much more discussion of why the endogenous switching model is producing this result.  (I am assuming that this is not the case, however, given the “Probit” implementation.  Please confirm.)

p. 6, line 237, beta_3 should also be included in this list.

4. Results and Discussion

The authors should define rho in the Materials and Methods section.  The value is interpreted in the first paragraph of results, but this is the first mention.  How does rho link to the equations on p. 5?

What is the interpretation of Rho0=1 in the last column of Table 4?  Is there something happening in terms of statistical power or specification for this model?  Is this result believable?

The format of Table 6 is hard to understand.  Please consider switching rows and columns and breaking into separate tables (perhaps split the categories into groups (first 3 in one table?  Second 3 in a separate table?)

5. Conclusions

Line 360, p. 10 The authors refer to gender, age, work engagement, and government subsidies as being important.  Are subsidies the only margin directly tested that can be changed by policy? 

The conclusions touch on this point but also add several other policy suggestions that seem less linked to the study that was actually conducted and the empirical tests that were actually done. 

Additional:

It would be nice to have a robustness section where the authors change the adoption definition away from the 0.5 cutoff to see if the major results change. 

It would also be nice to have robustness exercises without the presence of weighting via the coefficient of variation (table 1). 

Reviewer 2 Report

Comments and Suggestions for Authors

The work presented is very interesting, which reflects a scientific concern for the digitalization of the agricultural sector.

With the intention of improving the article, I consider that the following questions must be considered:

1. The authors have carried out 921 surveys, a high number, but considering that this data collection has been carried out in the areas with the largest cereal area in the country, it is worth asking: Is this a representative sample of all Chinese cereal producers? ? Can the results and statements made in the article be generalized to all cereal producers? If the study exceeds the number of surveys, it is an advantage for the authors.

2. By assigning the term “green” to an activity traditionally related to agriculture, the countryside, the agri-food chain, etc., it only leads to thinking that it is a fashionable issue, and therefore it is redundant. After reading the work, I think it refers more to "ecological production", since a change from a traditional productivity activity to a mode of production that respects the environment requires adopting technological innovations, with specific cultivation techniques, giving the research presented makes more sense, so I recommend: a change in the title.

3. The work proposes 7 hypotheses, of which only one has been confirmed, the rest, although it has been studied and analyzed, it is advisable to confirm, or not, whether it has been accepted or not. Indicate it in a section in the Results and discussion or Conclusions sections. With this, it provides more meaning to the set of hypotheses proposed.

4. Tables 3 and 4 contain a large amount of information that makes reading and interpretation difficult, which is why I would advise you to add a line between variables to integrate the elements that define them, as you have done in table 1. 

5. Finally, two equations are identified with the same number, (10).
